# Overcoming Spatial-Temporal Catastrophic Forgetting for Federated Class-Incremental Learning

## ABSTRACT

This paper delves into federated class-incremental learning (FCiL), where new classes appear continually or even privately to local clients. However, existing FCiL methods suffer from the problem of spatial-temporal catastrophic forgetting, i.e., forgetting the previously learned knowledge over time and the client-specific information owned by different clients. Additionally, private class and knowledge heterogeneity amongst local clients further exacerbate spatial-temporal forgetting, making FCiL challenging to apply. To address these issues, we propose Federated Class-specific Binary Classifier (FedCBC), an innovative approach to transferring and fusing knowledge across both temporal and spatial perspectives. FedCBC consists of two novel components: (1) continual personalization that distills previous knowledge from a global model to multiple local models, and (2) selective knowledge fusion that enhances knowledge integration of the same class from divergent clients and shares private knowledge with other clients. Extensive experiments using three newly-formulated metrics (termed GA, KRS, and KRT) demonstrate the effectiveness of the proposed approach.

## CCS CONCEPTS

• **Computing methodologies** → *Distributed algorithms*.

## KEYWORDS

Federated Continual Learning, Spatial-Temporal Catastrophic Forgetting

## 1 INTRODUCTION

Federated Continual Learning (FCL) is a novel yet non-trivial research topic that bridges Federated Learning (FL) and Continual Learning (CL), aiming at building a federated model to collaboratively learn a sequence (possibly never-ending) of tasks. Most existing FCL methods aim to integrate CL techniques into FL to enhance the practicality of FL [3, 11, 34, 36]. However, these methods suffer from a fundamental challenge, namely, *spatial-temporal catastrophic forgetting*. We will expatiate this challenge shortly.

In practice, the data collected by each client is usually different and heterogeneous [38]. For instance, a federated system of worldwide zoos would tackle different types of data. A desert wildlife monitoring station could have data about camels, and an oceanic

**Unpublished working draft. Not for distribution.**

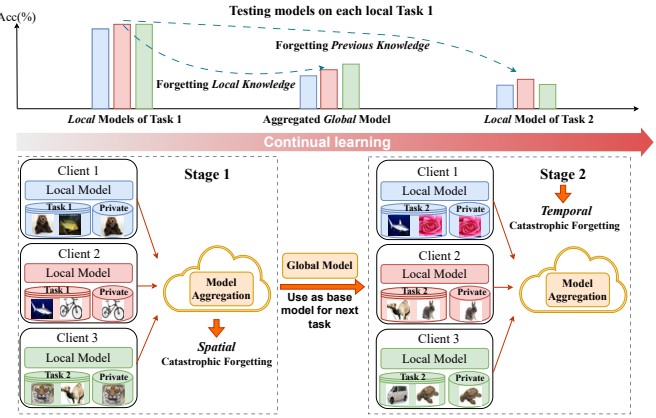

**Figure 1: An illustration example of *Federated Class-Incremental Learning* in real-world application, where each client needs to tackle a sequence of tasks, and each client has access to their private classes. For the pipeline, each client possesses a subset of private classes specific to itself. After completing the training on tasks with the same sequence number, the server will collect local models with knowledge of different classes and create a more powerful global model. Subsequently, the clients receive the global model and continue training on the remaining tasks based on the model they have received.**

monitoring station might have data about whales. We use the term "*private class*" to describe the classes that can only be accessed by one client. For us humans, we can gain indirect experience from others, acquiring knowledge that we haven't personally experienced. After learning the appearance of a tiger from the teacher, a child can easily identify it, even though he has never seen it before. It is seemingly easy for humans to adopt knowledge from others. However, it is extremely hard for a deep learning model to *fuse heterogeneous knowledge* from different clients training on different datasets, leading to severe spatial-temporal catastrophic forgetting [13, 27, 33].

What's more, knowledge acquired from others may conflict with one's existing knowledge [1]. Such a conflict will also happen in FCL. Human knowledge exchange occurs through communication. However, in FCL, knowledge exchange is accomplished through the aggregation of local models and the distribution of the global model. The knowledge of a model is often represented by parameters. Learning on heterogeneous data results in heterogeneous models. The more different the data, the greater the parameter divergence. Aggregating heterogeneous local models can lead to critical parameters for certain local tasks being overwritten, causing *spatial catastrophic forgetting*, i.e., poor performance on local

test sets [18, 19, 38]. When the aggregated global model is distributed to clients and used as a base model to continually learn new tasks, the important parameters of the previous task have been further rewritten [22], known as *temporal catastrophic forgetting*. Spatial-temporal catastrophic forgetting impedes traditional FL or CL techniques for FCL. The presence of private classes intensifies the heterogeneity among clients, making the fusion of local knowledge challenging and exacerbating spatial-temporal catastrophic forgetting [7, 39]. Existing research in FCL completely ignores the potential impacts of private classes. Moreover, they fail to solve spatial-temporal catastrophic forgetting, a fundamental challenge in FCL.

To thoroughly investigate the spatial-temporal catastrophic forgetting in FCiL under extremely imbalanced class distributions, we considered an real-world application setting, as illustrated in Fig. 1. In this problem, each client needs to continually tackle its training data, and the classes of these data also increase over time. Each client encounters different classes, including both public classes and private classes. Clients upload their direct experience to the server and receive indirect experience from other clients. In this way, clients continually gain the capability to recognize classes they have never observed before but are observed by other clients. The FCiL has two main objectives: the first is to ensure that local models do not forget the knowledge of old classes while learning new ones, which refers to overcoming temporal catastrophic forgetting. The second is to enable the global model to recognize all the classes encountered across all clients, addressing the problem of spatial catastrophic forgetting. To the best knowledge, we are the first to deal with such a problem.

To address these issues, in this work, we propose a novel framework called Federated Class-specific Binary Classifier (FedCBC), which effectively overcomes spatial-temporal catastrophic forgetting in FCiL. Generally, we introduce the concept of anomaly detection [32] for classification as a promising strategy. To be more specific, we construct a class-specific binary classifier for each class, rather than the conventional deep neural network approach. Moreover, our approach allows for the fusion of relevant knowledge while excluding conflicting knowledge. On the server side, we realize selective knowledge fusion, enhancing the generalization performance of the global model and mitigating spatial and temporal catastrophic forgetting. On the client side, we utilize the global model to generate previous data and add to the new task's dataset, thereby overcoming temporal catastrophic forgetting. Additionally, we employ the global model as a teacher model to perform knowledge distillation on the local model. Compared to several recent baseline methods, our approach achieves state-of-the-art performance in terms of average accuracy on various benchmark datasets. Moreover, we design three new metrics to evaluate the performance of models in this setting. The contributions of this paper are summarized as follows:

- We define a fundamental challenge in FCL, referred to as spatial-temporal catastrophic forgetting. In addition, we introduce a novel scenario, in which every client is required to perform class-incremental learning, and each client possesses private classes that are exclusively accessible to them, with no data from these classes ever being available to others.

- We propose a novel framework called FedCBC to address both spatial and temporal catastrophic forgetting. Moreover, employing variational autoencoders helps prevent the leakage of raw data, ensuring privacy and security. To our knowledge, the framework we designed exhibits state-of-the-art average accuracy performance in this problem domain.

- We design three new evaluation metrics in terms of global accuracy, spatial knowledge retention, and temporal knowledge retention to measure the degree of heterogeneous knowledge fusion and the level of spatial-temporal knowledge forgetting. Experimental results on three datasets show the superior performance of the proposed approach against baseline methods.

## 2 RELATED WORK

### 2.1 Federated Class Incremental Learning

Federated Class-Incremental Learning (FCiL) is a newly emerging research area, which focuses on overcoming catastrophic forgetting of previous tasks and data heterogeneity among clients jointly [12, 23, 37]. FedLwF [31] addresses catastrophic forgetting by distilling the knowledge of past local models to current local models, and addresses non-iid by distilling the general knowledge of global models to local models. GLFC [5] designs a class-aware gradient compensation loss to correct the imbalanced gradient propagation of old classes and a class-semantic relation distillation loss to keep inter-class relations consistent across tasks, and selects the best global model iteratively for preserving old knowledge with a proxy server. FedReconnaissance [11] treats the FCiL problem as maintaining the knowledge of the superset of classes observed by all clients and proposes to solve it with a prototypical network. AFCL [29] performs a prototype aggregation and a modified federated averaging aggregation on the server to overcome forgetting and client drift jointly.

However, existing methods overlook a crucial challenge in FCL, called spatial-temporal catastrophic forgetting. Additionally, they equally disregard the challenges posed by aggregating heterogeneous models when each client possesses unique private classes. Furthermore, these methods [5, 36] rely on storing a portion of old samples, thereby compromising privacy-preserving protocols.

### 2.2 Variational Auto-Encoder in FL

Variational auto-encoders refer to a class of generative models that aim to learn the probabilistic mapping between the data space and the representation latent space [14]. A typical VAE consists of two main components: an encoder and a decoder. The encoder maps input data into a probabilistic distribution in the latent space, while the decoder, on the other hand, maps samples from the latent space back to the data space [25]. The training objective of VAE is to minimize the reconstruction error while regularizing the latent space to follow a specific prior distribution. Instead of directly learning the conditional distribution $p(y|\mathbf{x})$, a VAE-based generative classifier learns the joint distribution $p(\mathbf{x}, y)$, which is factored as $p(\mathbf{x}|y)p(y)$, and to classify the samples via Bayes' rule [17]. In general, the application of VAE in federated learning is still limited, and existing works mainly focus on mitigating the cross-model covariate shift to address non-iid issues or detecting Byzantine attacks. VIRTUAL [2] uses a hierarchical Bayesian network on both

the client and server side, transfers posterior within the FL system, and performs interference with variational methods. FedDNA [6] decouples gradient parameters and statistical parameters to reduce the divergence between the global model and local models. FREPD [8] uses VAE to compute the reconstruction error of local updates to detect and defend against malicious attacks. ss In this paper, we innovatively combine VAEs with binary classifiers, utilizing the reconstruction loss of samples to serve as a class-specific binary classifier to mitigate the spatial forgetting of the global model, i.e., non-IID issues, and achieve continual personalization of local models. Continual personalization aims to ensure that, in the iterative federated training process, the clients do not underfit their private classes, even if private class samples are only accessed by the client itself and are non-dominant in quantity.

## 3 PROBLEM STATEMENT

### 3.1 Federated Class-Incremental learning

In traditional FL [20, 35], there are $a$ clients $\mathcal{A} = \{A_1, \ldots, A_a\}$ and one central server $S$. And each client $\{A_i, 1 \le i \le a\}$ only has access to its own data $\mathcal{D}_i$ due to privacy concerns. Basically, one communication round should contain three steps: 1) Server $S$ distributes the initial model or the global model from the last round to clients, 2) Client $A_i$ would use its private data $\mathcal{D}_i$ to train its local model $M_i$ based on the model from the server, and 3) Server collects local models $\{\theta_1, \ldots, \theta_a\}$ then aggregates them to update the global model. The performance of the final global model should be very close to the performance of a centralized trained model [24].

We now extend the traditional FL to the class-imbalanced FCiL.

- Given $a$ clients (denoted as $\mathcal{A} = \{A_1, A_2, \ldots, A_a\}$), and a central server (denoted as $S$), each client $\{A_i, 1 \le i \le a\}$ has its unique task sequence $\mathcal{T}_i$, which can differ significantly from one client to another. Suppose a set of public classes (denoted as $C_{pub}$) is accessible to all clients, and each client $A_i$ has its private class set $C_{pri}$. The primary objective of the local model $\theta_i$ is to incrementally learn to discriminate classes from the set $C_i = \{C_{pri} \cup C_{pub}\}$.

- The task sequence of client $A_i$ is denoted as $\mathcal{T}_i = \{T_i^1, T_i^2, \ldots, T_i^{n_i}\}$, where $n_i$ represents the total number of tasks on client $A_i$. The $k$-th task of $\mathcal{T}_i$ contains $\left|C_i^k\right|$ classes, and $C_i = \{C_i^1 \cup C_i^2 \cup \ldots, \cup C_i^{n_i}\}$.

- At task $r$, the global model $\theta_g^{r-1}$ can distinguish $\left|C_g^{r-1}\right|$ classes. The server $S$ then distributes it back to clients. Client $A_i$ uses $\theta_g^{r-1}$ as an initial model to train on its $r$-th task $T_i^r$. The local model $\theta_i^r$ should perform well in classifying classes from the set $\{C_g^{r-1} \cup C_i^r\}$.

- Finally, the server collects the local models from clients who participate in FCL and obtains a new global model $\theta_g^r$, which can identify classes from the set $C_g^r = \{C_g^{r-1} \cup C_1^r \cup C_2^r \cup \ldots \cup C_c^r\}$.

The goal of this setting is to end up with a global model that has assimilated all the tasks' knowledge acquired by individual clients, avoiding temporal catastrophic forgetting from the incremental local task progression and spatial catastrophic forgetting from aggregating heterogeneous local models of distinct clients (see Fig. 1).

## 3.2 Spatial-Temporal Catastrophic Forgetting

Catastrophic Forgetting is a fundamental challenge in CL, which mainly refers to a phenomenon that a model would forget the knowledge learned on old tasks when it is training on new tasks [4]. The reason for catastrophic forgetting is that the well-learned network parameters on the old tasks are overwritten during training on the new tasks [9]. In real-world applications, data is often collected gradually and a pre-trained model would continually train on the newly collected data for the new task requirements [28].

In the FCL setting, catastrophic forgetting exists as well. The assumption of static datasets in conventional FL is impractical. In a real-world scenario, data arrives at clients consecutively in the form of task streams, causing temporal catastrophic forgetting. When coming to the "aggregation" stage, the central server collects local models and aggregates them into one global model. After that, the server distributes the global model back to clients. Local models are trained with different training data. Aggregating them leads to the overwritten of certain task-specific crucial parameters, consequently causing a decline in the performance of the global model on local-specific tasks. Adopting the global model consolidated such conflict knowledge exacerbates the temporal catastrophic forgetting of each client itself previous tasks, especially for the non-overlapped classes, i.e., private classes.

In a nutshell, temporal catastrophic forgetting is caused by the unavailability of data in time. Spatial catastrophic forgetting is caused by the inaccessibility of data in space. In FCiL, clients also need to preserve the knowledge learned from previous tasks and learn new knowledge on newly arrived tasks. On the other hand, the server should achieve a selective knowledge fusion to maximize the retention of local knowledge from different clients, especially for the knowledge of those private classes.

## 4 PROPOSED METHOD: FEDCBC

In this section, we present the proposed method, i.e., Federated Class-specific Binary Classifier (FedCBC), to overcome spatial-temporal catastrophic forgetting, class privacy, and knowledge heterogeneity in FCiL. We firstly introduce binary classifiers for image classification instead of the traditional discriminative classifier in FCL. Specifically, on the client side, we construct a *Class-specific Binary Classifier* (**see Section 4.1**) for each class to determine whether a sample belongs to that class based on the reconstruction loss of variational auto-encoder. Subsequently, due to this unique network structure, it becomes easier to achieve *Selective Knowledge Fusion* (**see Section 4.2**) at the server, avoiding spatial catastrophic forgetting. It also enables the knowledge of private classes to be shared seamlessly between the server and clients. Finally, after receiving the more generalized global model from the server, the clients proceed to perform *Continual Personalization* (**see Section 4.3**) locally. This adaptation of the global model to local data distributions helps prevent temporal catastrophic forgetting.

The overall framework of the proposed method is shown in Fig. 2 and the algorithm is summarized in Algorithm 1.

### 4.1 Class-specific Binary Classifier

When learning on a new task, the parameters of the network trained on previous tasks are overwritten, which results in temporal

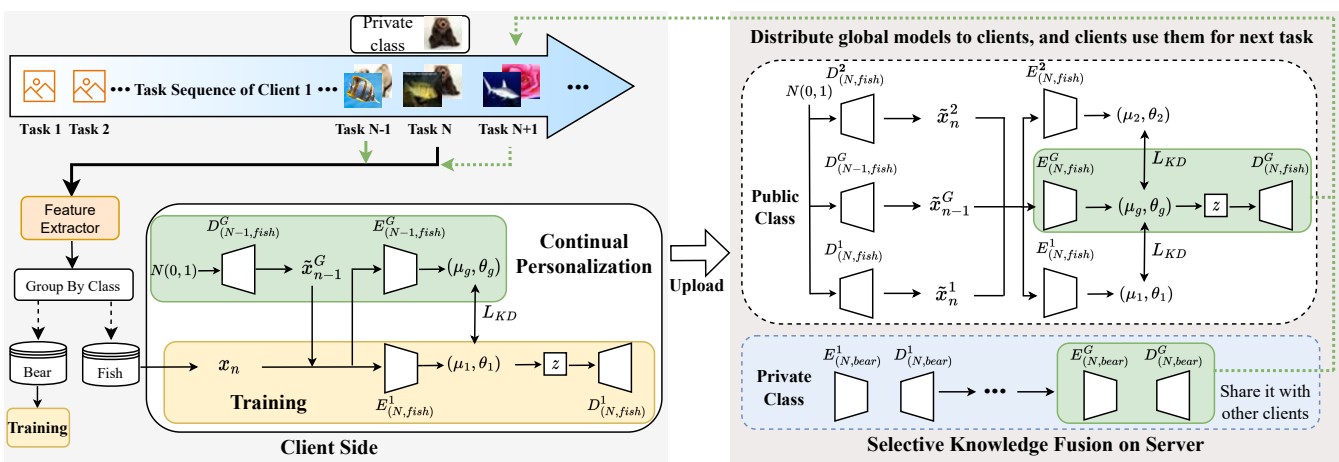

Figure 2: An overview of the proposed FedCBC. Class-specific BCs are adopted to avoid temporal forgetting caused by learning new classes. On the server, Selective Knowledge Fusion fuses knowledge of the same class from different clients, avoiding spatial catastrophic forgetting due to the fusion of unrelated knowledge. It can also alleviate temporal catastrophic forgetting by incorporating the global model from the last task into the process. On the client, Continual Personalization allows general knowledge adapted to the local distribution, avoiding temporal catastrophic forgetting.

catastrophic forgetting. Furthermore, the overwritten parameters also bring about an overfitting towards new classes. In a vanilla class-incremental setting, the norms of weight vectors in the full-connected layer for new classes tend to be larger, leading to network mispredictions of samples from previous classes as new ones.

Based on the above analysis, it can be concluded that the key to mitigating forgetting is to prevent interference among class knowledge. To address this issue, we propose to use class-specific VAEs to memorize the specialized knowledge of each class separately. We start by training class-specific VAEs for each class on each client, and preserve the class-wise knowledge in them. When we compute the reconstruction loss with the original samples (features), they become a kind of class-specific binary classifier. When encountering a new class, all we need is to train another classifier for the new class, leaving the rest for previous classes unchanged.

**Training Stage:** A VAE model consists of two parts: an encoder $q_\phi$ and a decoder $p_\theta$. The encoder $q_\phi$ maps the input $x$ to a posterior distribution $q_\phi(z|x)$, and the decoder $p_\theta$ is used to reconstruct the input sample $x$ from the latent variable $z$. Moreover, the prior distribution $p_{prior}(z)$ is typically assumed to follow a standard normal distribution during the training process of a VAE, which is defined as:

$$q_\phi(z|x) = \mathcal{N}(z|\mu_\phi^{(x)}, \sigma_\phi^{(x)^2}),  \quad (1)$$

$$p_\theta(x|z) = \mathcal{N}(x|\mu_\theta^{(z)}, 1),  \quad (2)$$

$$p_{prior}(z) = \mathcal{N}(0, 1),  \quad (3)$$

where $\mu_\phi^{(x)}$ and $\sigma_\phi^{(x)^2}$ are the output of the encoder, and $\mu_\theta^{(z)}$ is the output of the decoder.

The VAE models are trained by optimizing two parts. The first is about the Kullback-Leibler divergence between $q_\phi(z|x)$ and $\mathcal{N}(0,1)$. The second part is the reconstruction loss. Formally, the lower

bound (or ELBO) is formulated as follows

$$\mathcal{L}_{\text{ELBO}}(\theta, \phi; x) = E_{q_\phi(z|x)}\left[\log \frac{p_\theta(x, z)}{q_\phi(z \mid x)}\right]$$
$$= E_{q_\phi(z|x)}\left[\log p_\theta(x \mid z)\right] - D_{KL}\left(q_\phi(z \mid x) \| p_{\text{prior}}(z)\right). \quad (4)$$

Therefore, the whole training stage can be summarized as a loss function:

$$\mathcal{L}_{loss} = (1-\alpha)*D_{KL}(\mathcal{N}(\mu_\phi^{(x)}, \sigma_\phi^{(x)^2})|\mathcal{N}(0,1))+\alpha*MSE(x,x'). \quad (5)$$

**Prediction Stage:** After the training stage, each client has a set of VAEs containing class-specific knowledge. The class-specific knowledge is stored locally in the form of key-value pairs, where the key is the class name and the value is VAE.

When it comes to the prediction stage, the test sample $x$ would go through all the VAEs and generate $v$ samples, where $v$ represents the number of VAEs. And the one that has the smallest reconstruction loss is the prediction class. In summary, the classification is done using:

$$\hat{y}^x = \arg\min_{y \in C} MSE(x, x'_y), \quad (6)$$

where $C$ denotes the entire class set and $x'_y$ is the reconstruct sample from the VAE of class $y$.

## 4.2 Selective Knowledge Fusion

Spatial catastrophic forgetting is caused by the aggregation of heterogeneous local models. Due to variations in data distribution and classes, each local model acquires distinct knowledge. Simultaneously, the inexplicability of deep neural networks results in the entanglement of class-specific knowledge within the network, making it difficult to isolate individual class knowledge. This results in the overwriting of knowledge among classes within aggregate models, subsequently causing the merged global model to perform below expectations on local datasets.

**Algorithm 1:** Proposed FedCBC Algorithm

**Input:** $a$ clients $\mathcal{A} = \{A_i\}_{i=1}^a$ with their own task sequence $\mathcal{T}_i = \{T_i^n\}_{n=1}^N$.

**Output:** Global models in the form of $M_g^N = \{$Class $C_g$: model $\theta_{(g,C_g)}^N \}$.

1 Initialization;
2 **while** *task number* $n \leq N$ **do**
3    **for** *each client $A_i$, $1 \leq i \leq a$* **do**
4      $\{C_i^n \colon \mathcal{D}_C^n\} \leftarrow$ GroupByClassLabel($T_i^n$);
5      **for** *class $c \in C_i^{T_i^n}$* **do**
6        **if** $\theta_{(g,c)}^{n-1}$ *is in* $M_g^{n-1}$ **then**
7          $\theta_{(i,c)}^n \leftarrow$ ContPers($\theta_{(g,c)}^{n-1}, \mathcal{D}_c^n$);
8        **else**
9          $\theta_{(i,c)}^n \leftarrow$ TrainLocalModel($\mathcal{D}_c^n$);
10    **Server aggregation:**
11    $C_g^n = \{C_1^{T_1^n} \cup \ldots \cup C_a^{T_a^n} \}$;
     // Set of classes seen by all clients in the $n$-th task
12    **for** *class $c \in C_g^n$* **do**
13      set an empty local model list $M_c$ of class $c$;
14      **if** $\theta_{(g,c)}^{n-1}$ *is in* $M_g^{n-1}$ **then**
15        add $\theta_{(g,c)}^{n-1}$ into $M_c$;
16      **if** *client $A_i$ has a model of $c$* **then**
17        add $\theta_{(i,c)}^n$ into $M_c$;
18      $\theta_{(g,c)}^n \leftarrow$ SelectiveKnowledgeFusion($M_c$);
19      add $\theta_{(g,c)}^n$ into $M_g^n$;
20    Distribute $M_g^n$ to all clients.

In order to mitigate temporal-spatial forgetting, we introduced selective knowledge fusion on the server side. The main idea is to merge knowledge about the same class from different clients, along with the integration of past knowledge. Since the class-wise knowledge is stored separately in different VAEs, the selective knowledge fusion process is straightforward by simply consolidating the lists of key-value pairs uploaded from clients. Specifically, the server will group the model key-value pairs collected from various clients based on their class names. That is, the VAEs of the same class while from different clients are grouped together and fused separately. Such an approach by preventing the merging of unrelated knowledge is useful to avoid spatial catastrophic forgetting, especially when the data distribution is extremely Non-IID. Furthermore, if there is already a global model of the same class from the previous round, it can be also included in the group for selective knowledge integration. It is still helpful for alleviating temporal catastrophic forgetting.

In the FCiL setting, the client's dataset consists of two types of data: *samples from public classes* and *samples from private classes*. Private classes refer to those classes that only the respective client has access to throughout the entire training process. Aggregation enables clients to acquire knowledge about private classes from other clients, granting them the capability to identify classes they have never encountered before. This approach avoids direct data sharing and prevents privacy.

**Public Class:** Public class implies that multiple clients possess training data for this class and upload related models. Therefore, within this group, there will be multiple class-specific VAEs, including the global model of the last round if it exists. In such groups, we feed Gaussian noise data sampled from a normal distribution into the decoder part of the VAE to generate pseudo-samples. While these pseudo-samples belong to the same class, each VAE generates samples with its own unique local characteristics, much like coffee beans from different origins.

Subsequently, these pseudo-samples are used as a training set for the next distillation step to generate a more generalized global model. First, we initialize a new VAE as the global model of this class, denoted as $M_g$. For the decoder part of $M_g$, the traditional MSE loss is used to train its reconstruction ability. And for the encoder part, there's something different about the training process. We consider the other local VAEs as teacher models, while the global VAE is the student model. The encoder of $M_g$ maps input $x$ to a posterior distribution $q_\phi^g(z|x) = \mathcal{N}(\mu_g, \sigma_g^2)$. Training the encoder part involves reducing both the KL divergence between $\mathcal{N}(\mu_g, \sigma_g^2)$ and $\mathcal{N}(0, 1)$ and the KL divergence between $\mathcal{N}(\mu_g, \sigma_g^2)$ and the average posterior distributions of other VAEs $\mathcal{N}(\bar{\mu}, \bar{\sigma}^2)$. The final loss function of this stage is formulated as:

$$
\mathcal{L}_{kd} = \alpha * MSE(x, x') + \beta * D_{KL}(\mathcal{N}(\mu_g, \sigma_g^2)|\mathcal{N}(0,1)) \\
+ (1 - \alpha - \beta) * D_{KL}(\mathcal{N}(\mu_g, \sigma_g^2)|\mathcal{N}(\bar{\mu}, \bar{\sigma}^2)) \tag{7}
$$

where $\alpha$ and $\beta$ are the hyperparameters. Finally, the generalized global VAE is obtained.

**Private Class:** Private class means that there is only one local VAE specific to that class within the group. A naive method for private classes is to use this model directly as the global model. However, due to privacy concerns, this method is not feasible. Therefore, we follow a similar approach for handling public classes, where the local models and the previous round's global model still rehearsal samples. Afterward, these pseudo-samples are used as the training set to train and distillate the global model. Once the global model for a private class is trained, it will be distributed to the participating clients along with the models for other public classes. This way, other clients gain the capability to identify classes they have never encountered before, achieving knowledge sharing.

## 4.3 Continual Personalization

Following the selective knowledge fusion on the server side, all global VAEs are distributed to the clients that just participated in the aggregation process. For the classes they are familiar with, clients will possess a more generalized VAE. Simultaneously, for classes that they have not encountered themselves but others have, clients will also have the capability to recognize them, as they have received knowledge about these unknown classes from others.

Although the global model would be more generalized, the performance on the local test set could still be worse than existing local models [26]. Moreover, when learning new data about the old classes, some knowledge may still be forgotten because of concept

drift. Therefore, based on the idea of personalized FL [30], the global model would only be used as a teacher model on the client side. On the one hand, the global model rehearsals previous knowledge samples and integrates them into newly collected data, thereby curbing temporal catastrophic forgetting at the data level. On the other hand, employing knowledge distillation limits the output of local models, mitigating temporal catastrophic forgetting in terms of model output.

## 5 EXPERIMENTS

### 5.1 Experiment Setup

**Datasets.** To evaluate the performance of our method, we use three datasets: MNIST [16], CIFAR-10 [15] and CIFAR-100 [15] in our experiments. In our setup, the federated system consists of three clients and one central server, and each client possesses a sequence of five unique tasks. Initially, we divide the data for each class into three parts using a Dirichlet distribution to ensure that there is no data overlap between clients. For MNIST and CIFAR-10, each client exclusively owns two classes that were only accessible to itself and not accessible to others. Therefore, all clients could only access four classes. The data for each client's tasks is randomly sampled from these six classes, with three classes chosen for each task. For CIFAR-100, we allow each client to have 25 private classes, resulting in 25 common classes. Each task consists of 10 classes sampled from both private and common classes, with no class overlap between tasks.

**Baselines.** To have a comprehensive evaluation, we compared our method with representative existing FCiL methods. The compared methods include: (1) **FedAvg** [24], a standard approach for FL. (2) **FedAvg+EWC**, integrating a regularization-based approach of continual learning to the standard FL framework. (3) **FedProx** [21], a well-known FL method aiming to address statistical heterogeneity. (4) **GLFC** [5], a newest and famous FCiL method using multiple complex components. (5) **FedSpace** [29], an asynchronous FCiL method. For baseline algorithms, we employed ResNet-18 [10] as the backbone network.

**Implementation.** The code of each method is implemented in PyTorch. For the baseline algorithms, the employed classification network is ResNet-18. We conducted experiments on each dataset using three different random seeds (42, 1999, 2002) and averaged the results. We set the number of global epochs as 5 and the number of local epochs as 50. For CIFAR-10 and CIFAR-100, we utilized 5% of the data of each class for pretraining the feature extractor. The whole training process is performed sequentially on an NVIDIA GPU RTX-3090. Our code is now anonymously hosted at: https://anonymous.4open.science/r/FedAE-CDE7/.

### 5.2 Evaluation Metrics

Since spatial-temporal catastrophic forgetting is a novel challenge that we first introduced, lacking measurements, we have designed three different metrics to assess heterogeneous knowledge integration, temporal knowledge retention and spatial knowledge retention. All three metrics are designed based on accuracy.

**Global accuracy.** Specifically, it is the accuracy of the global model testing on the test set of all classes. This metric is used to measure the degree of heterogeneous knowledge fusion. Since each

client has some private classes, testing the aggregated global model on a test set containing all classes can detect whether these unique knowledge aspects have been preserved. For example, if only one client's model was trained on Apple images, and after aggregation, the global model still performs well on an Apple test set, it indicates that it has retained the unique knowledge about apples without being overwritten. In short, it is used to evaluate the ability of the global model to recognize all the classes encountered across all clients.

**Temporal knowledge retention.** We use *Knowledge Retention* as measurement of forgetting. Temporal knowledge retention is designed to measure the extent to which local models retain knowledge of old tasks as they learn on the task sequence. $Acc_i^{(0,0)}$ represents the accuracy of client $i$'s local model trained on the first task testing on the test set of the first task. And $Acc_i^{(r,0)}$ represents the accuracy of client $i$'s local model trained on the $r$-th task testing on the test set of the first task. The ratio of these two values provides insight into how much knowledge the local model retains from the first task when it has completed training on the $r$-th task. Therefore, the spatial catastrophic forgetting can be expressed in equation 8.

$$KR_t = \frac{1}{N} \sum_{i=1}^{N} \frac{Acc_i^{(r,0)}}{Acc_i^{(0,0)}} \qquad (8)$$

where $N$ represents the number of clients.

**Spatial knowledge retention.** Similarly, we can deduce the expression form of spatial catastrophic forgetting in equation 9. This metric is designed to measure how much local-specific knowledge is retained by the aggregated global model. A smaller value indicates that more local knowledge was overwritten during aggregation.

$$KR_s = \frac{1}{N} \sum_{i=1}^{N} \frac{Acc_g^{(r,r_i)}}{Acc_i^{(r,r)}} \qquad (9)$$

where $Acc_g^{(r,r_i)}$ is the accuracy of the global models on the $r$-th testset of client $i$. And the global model is obtained by aggregating the local models trained on the $r$-th task from all the clients.

### 5.3 Experimental Results

**Table 1: Average global accuracy on MNIST with 5 class-incremental tasks each client.**

| Algorithm | Task ID | | | | | |
|---|---|---|---|---|---|---|
| | 1 | 2 | 3 | 4 | 5 | Avg. |
| FedAvg[24] | 16.38 | 27.25 | 29.29 | 29.78 | 23.69 | 25.28 |
| FedAvg+EWC | 10.06 | 9.53 | 10.30 | 10.06 | 10.08 | 10.01 |
| FedProx[21] | 14.81 | 10.57 | 14.99 | 13.08 | 10.22 | 12.73 |
| GLFC[5] | 53.56 | 55.99 | 51.24 | 65.66 | 51.46 | 55.58 |
| FedSpace[29] | 25.32 | 26.17 | 31.48 | 34.27 | 37.26 | 30.90 |
| Ours (FedCBC) | **64.90** | **77.17** | **85.86** | **87.46** | **90.01** | **81.08** |

In Table 1 to Table 3, we reported the accuracy after 5 global epoch training each task and compared the performance with GLFC, AFCL, FedSpace, FedProx, FedAvg and FedAvg+EWC. Under the challenging restriction of federated private class incremental setup,

**Table 2: Average global accuracy on CIFAR-10 with 5 class-incremental tasks each client.**

| Algorithm | Task ID | | | | | |
| --- | --- | --- | --- | --- | --- | --- |
| | 1 | 2 | 3 | 4 | 5 | Avg. |
| FedAvg[24] | 20.25 | 16.71 | 24.57 | 24.29 | 23.79 | 21.92 |
| FedAvg+EWC | 10.00 | 10.00 | 10.00 | 9.78 | 10.03 | 9.96 |
| FedProx[21] | 16.27 | 10.18 | 10.39 | 12.33 | 12.64 | 12.36 |
| GLFC[5] | 41.55 | 43.59 | 38.60 | 44.54 | 45.02 | 42.66 |
| FedSpace[29] | 23.56 | 22.05 | 25.63 | 25.93 | 25.50 | 24.53 |
| Ours (FedCBC) | **45.94** | **55.76** | **58.86** | **61.36** | **67.74** | **57.93** |

**Table 3: Average global accuracy on CIFAR-100 with 5 class-incremental tasks each client.**

| Algorithm | Task ID | | | | | |
| --- | --- | --- | --- | --- | --- | --- |
| | 1 | 2 | 3 | 4 | 5 | Avg. |
| FedAvg[24] | 1.45 | 1.52 | 1.63 | 1.67 | 1.28 | 1.51 |
| FedAvg+EWC | 0.86 | 1.00 | 1.00 | 1.00 | 1.00 | 0.97 |
| FedProx[21] | 1.39 | 1.00 | 1.00 | 1.00 | 1.03 | 1.08 |
| GLFC[5] | 9.27 | 9.91 | 11.37 | 10.63 | 10.97 | 10.43 |
| FedSpace[29] | 4.30 | 4.68 | 5.34 | 4.53 | 4.36 | 4.64 |
| Ours(FedCBC) | **13.24** | **19.17** | **23.35** | **26.48** | **29.35** | **22.32** |

FedAvg+EWC failed and showed the poorest performance on all datasets, we believe that is due to the inapplicability of the EWC method in class-incremental scenarios is the root cause. FedProx was also below expectations. While it is used to address statistical heterogeneity, obviously it cannot handle such a challenging problem. Although FedSpace and GLFC have made special provisions for such highly Non-IID scenarios, experimental results indicate that they still struggle to effectively integrate the knowledge of heterogeneous local models.

From the results, we can see our method achieves 90.01% on MNIST, 67.74% on CIFAR-10, and 29.35% on CIFAR-100, showing the state-of-the-art performance of fusing heterogeneous local models.

In the following section (i.e., Sec. 5.4), we will evaluate each method using new metrics, i.e., temporal knowledge retention and spatial knowledge retention, to evaluate the resistance of spatial-temporal catastrophic forgetting.

### 5.4 Ablation Studies

To validate the effectiveness of our proposed method in mitigating both spatial and temporal catastrophic forgetting, we conducted experiments along with baselines to test the preservation of spatial-temporal knowledge. Additionally, we performed ablation experiments on our method. Fig. 3 shows the spatial knowledge retention of all methods on three datasets. In Fig. 3b and Fig. 3c, we notice that after aggregation on the server, the global model performs even better than the local models. It indicates our method is robust to spatial catastrophic forgetting. However, the spatial knowledge retention drops shapely when we remove the selective knowledge fusion on the server side (*Ours-w/oSKF*). It indicates this strategy helps.

Fig. 4 shows the temporal knowledge retention of all methods, which indicates the ability to migrate catastrophic forgetting in time. When we removed the continual personalization on the local side, the temporal knowledge retention would drop to around 90% on the MNIST and CIFAR10. It is mainly determined by the model's architecture and is not heavily influenced by the mechanisms.

### 5.5 Communication Cost Analysis

**Table 4: The Number of Parameters in different backbone networks.**

| BackBone | Number of Trainable Parameters |
| --- | --- |
| ResNet-18 | 11,306,804 |
| Binary Classifier | 361,728 |

Tab. 4 illustrates the number of parameters that need to be trained in two different backbone networks. Compared to a ResNet-18 with 11,306,804 parameters, a single VAE only has 361,728 parameters (in our experiment setting). So training a VAE is much less challenging than a ResNet-18.

Furthermore, we set up one VAE module for each class. However, due to the redundancy of neural networks, data of multiple classes can be included in a single VAE module for classification, further reducing communication overhead and storage space.

Although the storage space required by our method grows linearly with the number of classes when facing a large number of class-labeled data, the increase in parameter count is tolerable compared to its superior performance in overcoming catastrophic forgetting in both spatial and temporal aspects.

### 5.6 Sensitivity & Privacy Analysis

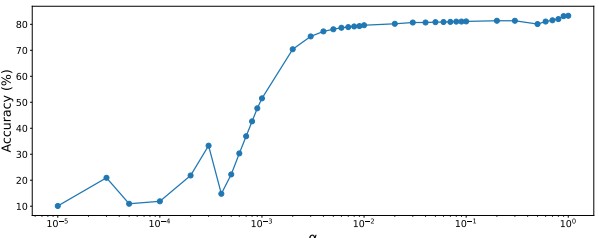

**Figure 5: As $\alpha$ increases, the quality of replayed fake samples improves, enhancing the accuracy of the method but simultaneously reducing the security of privacy.**

Fig. 5 shows the accuracy of our method as $\alpha$ in Equ. 7varies. On one hand, $\alpha$ controls the quality of generated pseudo-samples by regulating the weight of the MSE loss. On the other hand, $1 - \alpha$ controls the weight of the KL divergence between the true latent distribution and the standard normal distribution. In other words, higher $\alpha$ values indicate higher quality of generated pseudo-samples and lower privacy protection. Conversely, lower $\alpha$ values signify a more distorted shift in the latent distribution, resulting in stronger privacy protection.

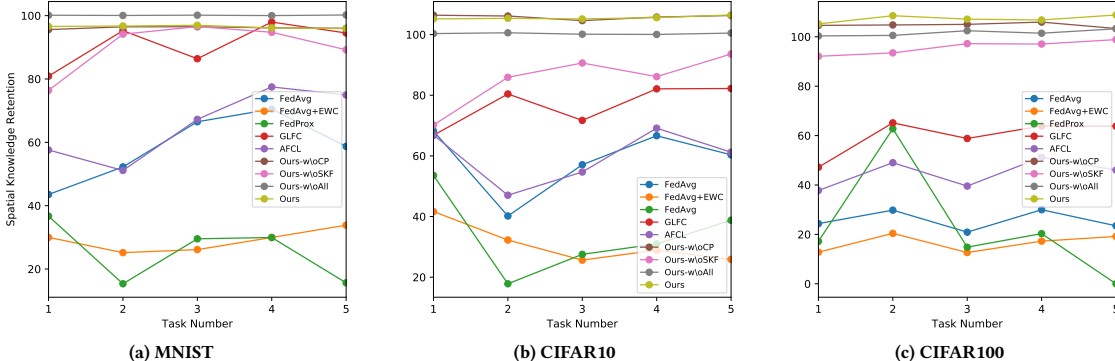

(a) MNIST

(b) CIFAR10

(c) CIFAR100

Figure 3: Spatial knowledge retention (Eq. 9). Note that, 'Ours-w/oSKF' refers to our method without selective knowledge fusion on the server side. 'Ours-w/oCP' refers to our method without continual personalization on the client side.

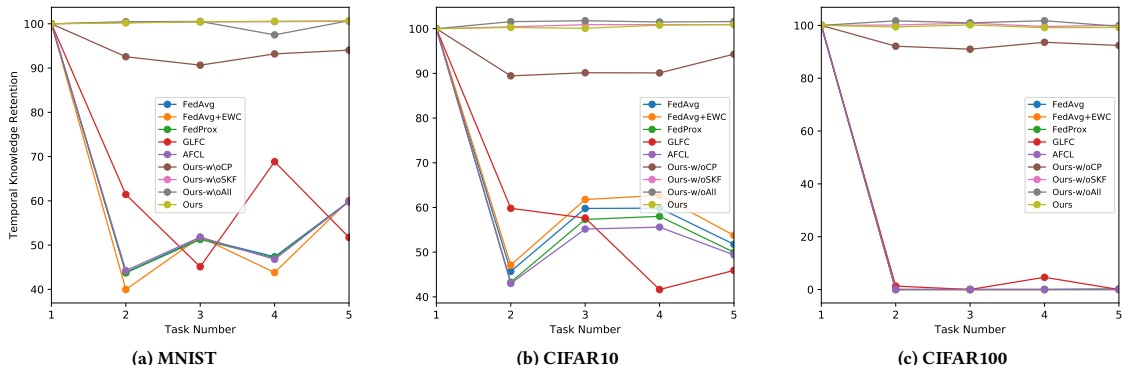

(a) MNIST

(b) CIFAR10

(c) CIFAR100

Figure 4: Temporal knowledge retention (Eq. 8).



Figure 6: Visualization of the pseudo-samples generated by our method on MNIST. It demonstrates that our approach can control the quality of the pseudo-samples by adjusting the value of $\alpha$, thereby balancing the accuracy and privacy.

Fig. 6 shows the visualization of the generated pseudo-samples on MNIST when $\alpha = 10^{-3}$ and $\alpha = 10^0$. Clearly, when $\alpha = 10^{-3}$, the generated pseudo-samples are very blurry, resulting in the lowest accuracy for FedCBC. However, when $\alpha = 10^0$, although the generated pseudo-samples are clear and the accuracy is satisfactory, privacy protection is not controlled. FedCBC can adjust the trade-off between privacy security and performance by changing $\alpha$

## 6 CONCLUSION

Federated Clsss-Incremental Learning (FCiL) is a novel yet non-trivial research topic. This paper investigated a new and real-word setting problem, where new classes appear continually to each client and some classes are private to certain clients. Therefore, class privacy emerging on certain clients and knowledge heterogeneity coming from different clients are two basic challenges for this problem. In addition, we discussed a significant challenge, referred to as Spatial-Temporal Catastrophic Forgetting.

To address these challenges, we proposed a Federated Class-specific Binary Classifier (FedCBC) approach. To evaluate the performance of FedCBC and its ability to resist spatial-temporal catastrophic forgetting, we designed three new metrics to measure the ability to fuse heterogeneous knowledge and the preservation of temporal and spatial knowledge. Experimental results on three datasets showed that the proposed approach outperformed the existing baseline methods markedly.

We are interested in its potential to inspire future research in this domain. Our future work includes: 1) exploring more effective technologies for heterogeneous knowledge fusion on the server side, 2) considering additional constraints in FL, such as fairness and robustness, and 3) further devising a holistic method to tackle heterogeneous FCiL settings.

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
