# OpenReview forum: "Overcoming Spatial-Temporal Catastrophic Forgetting for Federated Class-Incremental Learning"
_acmmm.org/ACMMM/2024/Conference — MM2024 Poster_

### Official Review · Reviewer_Bvrf · 2024-05-18

**Rating:** 4
**Confidence:** 4

**Summary:**

This paper addresses the challenge of Spatial-Temporal Catastrophic Forgetting in Federated Class-Incremental Learning, a critical concern for maintaining continuous learning without losing previously gained knowledge in distributed systems. To counter this issue, the paper introduces the Federated Class-specific Binary Classifier (FedCBC), a ground-breaking framework designed to mitigate forgetting through two key innovations: (1) continual personalization, which distils and transfers knowledge from a global model to local models, and (2) selective knowledge fusion, which integrates and shares class-specific insights among diverse clients. The efficacy of FedCBC is evaluated using three newly developed metrics: GA, KRS, and KRT. Experimental results demonstrate that FedCBC not only improves learning accuracy but also reduces communication costs compared to five other existing methods.

**Strengths:**

This paper includes two novelties: The introduction of the FedCBC framework as a novel solution to the issue of Spatial-Temporal Catastrophic Forgetting in Federated Class-Incremental Learning is a primary novelty. The introduction of three new metrics (GA, KRS, KRT) for evaluating the performance of federated learning models is another strength, which provide a broader and perhaps more precise evaluation framework tailored to the specific challenges addressed by the paper.

The approach replaces the traditional discriminative classifier with binary classifiers, each specific to a class, which evaluate if a sample belongs to a class based on the reconstruction loss from a variational auto-encoder. This setup simplifies the Selective Knowledge Fusion process on the server side, effectively preventing spatial catastrophic forgetting and facilitating the sharing of knowledge about private classes between the server and clients. Following the server's distribution of a more generalized global model, clients customize this model locally through Continual Personalization to adapt to local data variations, thereby mitigating temporal catastrophic forgetting.

The paper makes extensive experiments, which compare FedCBC against five existing methods using three datasets, lend significant credibility. This comparative analysis underscores FedCBC's improvements in terms of accuracy and communication costs.

In summary, the paper’s introduction of the FedCBC framework is notable for its innovative approach to a significant problem in federated learning. The theoretical underpinnings are sound, and the detailed experimental evaluation provides a strong justification for the proposed methods. The clarity of the presentation enhances its impact, making it a valuable contribution to the field.

**Limitations:**

In section 1, the explanation of the causes of spatial-temporal catastrophic forgetting is somewhat dispersed and could benefit from being more logical and clearer. For example, Line 101-104 and line 130 need more specific explanation and evidence to help reader understand well.

In section 3.2, comparing to FL and FCL, the analysis of what leads to spatial-temporal catastrophic forgetting in FCiL is inadequate.

In section 5.3 and 5.4, There are inconsistencies of components between table 1-3 and figure 3-4, which affect the credibility and persuasiveness of experimental data and results. AFCL is not in the table but in the figure. FedSpace is in the table but not in the figure. If this is correct, it should be stated clearly at the beginning of 5.3 and 5.4.

Other minor errors: Line 131 ‘an’ should be ‘a’, line 658 ‘spatial’ may be ‘temporal’.

**Suitability:**

2

---

### Official Review · Reviewer_keep · 2024-05-19

**Rating:** 5
**Confidence:** 4

**Summary:**

This paper focuses on federated class-incremental learning and points out the spatial-temporal catastrophic forgetting including forgetting previous knowledge and forgetting local knowledge.  The author introduces the Federated Class-specific Binary Classifier and demonstrate the effectiveness of the proposed method.

**Strengths:**

The paper is well-written and easy to understand. The problems are clear with the respective solutions.

**Limitations:**

There are several limitations.

First, the author introduces the class-specific VAEs for each class on each client. However, it brings additional computation costs for each client and requires additional communication costs in the federation. I urge the authors to discuss the relative computation and communication cost with the relative methods.

Second, I recommend adding the experiments with more task-splitting settings. For example, CIFAR-100  with 10/20 lass-
incremental tasks for each client. Furthermore, I wonder whether to conduct the experiments with various feature shifts. For example, digits setting, different client from the MNIST, SVHN domains to evaluate the robustness of VAE.

**Suitability:**

2

---

### Official Review · Reviewer_5S9j · 2024-05-23

**Rating:** 4
**Confidence:** 4

**Summary:**

This manuscript proposes a federated class-specific binary classifier framework, named FedCBC, to address the issue of spatial-temporal catastrophic forgetting in federated class-incremental learning (FCiL). FedCBC transfers and fuses knowledge across both temporal and spatial perspectives through continual personalization and selective knowledge fusion. Experiments have been conducted on three datasets.

**Strengths:**

1. **Novelty and Importance:** This manuscript is the first study to address the spatial-temporal catastrophic forgetting problem in FCiL, which is crucial and fundamental. In this scenario, each client possesses private classes, with no data from these classes being available to other clients.
2. **Innovative Framework:** The proposed framework, FedCBC, is presented clearly and contributes several new ideas. The main ideas of FedCBC are creative and distinctive.
3. **Comprehensive Experiments:** Many experiments have been conducted to validate the effectiveness of the proposed method.

**Limitations:**

1. **Baselines:** The chosen baselines are not strong enough. FedSpace is published in a related workshop. Additionally, only GLFC is specifically designed for federated class-incremental learning. Please provide more comparison results with other previous works.
2. **System Complexity:** The use of variational autoencoders (VAEs) may increase the system's complexity and computational overhead. Thus, the training time of FedCBC compared to other baseline methods needs to be included.
3. **Privacy Analysis:** The paper does not mention the theoretical analysis of privacy protection provided by VAEs, despite claims that they help prevent the leakage of raw data.

**Suitability:**

1

---

### Official Review · Reviewer_5UNr · 2024-05-27

**Rating:** 5
**Confidence:** 3

**Summary:**

This paper proposes FedCBC to overcome Spatial-Temporal Catastrophic Forgetting on class-incremental learning.

**Strengths:**

1. The methods are intuitive and well-motivated.
2. The figures are good.
3. Experiments are extensive and solid.

**Limitations:**

1. Comparsions are not sufficient. More SOTA Federated methods should be included, not just FedProx.
2. The concept of Spatial-Temporal Catastrophic Forgetting have not been sufficiently analyzed, remaining on the words description. Reference:
Lee G, Jeong M, Shin Y, et al. Preservation of the global knowledge by not-true distillation in federated learning[J]. Advances in Neural Information Processing Systems, 2022, 35: 38461-38474.
Chen J, Zhu J, Zheng Q. Towards Fast and Stable Federated Learning: Confronting Heterogeneity via Knowledge Anchor[C]//Proceedings of the 31st ACM International Conference on Multimedia. 2023: 8697-8706.

**Suitability:**

2

---

### Meta-Review · Area_Chair_NECp · 2024-06-29

**Recommendation:** Accept (Poster)
**Confidence:** 3

**Metareview:**

This paper focuses on federated class-incremental learning and points out the spatial-temporal catastrophic forgetting including forgetting previous knowledge and forgetting local knowledge. The author introduces the Federated Class-specific Binary Classifier with continual personalisation and selective knowledge fusion. The authors also suggest evaluation strategies for the efficacy of the proposed methods, of which the results reveal the superiority of their proposal. All the reviewers agree that the paper has some novelty and the experimental results are sound. However, it requires additional training time and more efforts should be paid to explain the concept of spatial-temporal catastrophic forgetting. Overall, I tend to accept the paper, while I hope the authors could pay more efforts to address aforementioned issues in their revision.

---

### Meta-Review · Senior_Area_Chairs · 2024-07-10

**Recommendation:** Accept (Poster)
**Confidence:** 4

**Metareview:**

All the reviewers gave positive ratings and tend to accept the paper. SAC and AC agree with reviewers and recommend acceptance of the paper.